# Effect of B on Microstructure and Properties of Surfacing Layer of Austenitic Stainless Steel Flux Cored Wire

**DOI:** 10.3390/ma15175884

**Published:** 2022-08-26

**Authors:** Jianbo Guo, Zhengjun Liu, Yunhai Su

**Affiliations:** School of Materials Science and Engineering, Shenyang University of Technology, Shenyang 110870, China

**Keywords:** boron element, welding, microstructure, corrosion resistance, electrochemistry

## Abstract

In order to study the effect of element B on the corrosion resistance of stainless steel-based flux cored wire surfacing alloy, a stainless steel surfacing layer was prepared on the surface of carbon steel plate by melt electrode gas shielded welding, and then the microstructure, electrochemical corrosion resistance, and wear resistance of the surfacing layer were analyzed. The results show that the surfacing layer of surfacing alloy presents M_2_B and Fe_3_(C, B) phases based on austenite. Boride formed in deposited metal has good corrosion resistance. Therefore, adding the proper amount of B can significantly improve the corrosion resistance of the surfacing layer. When the boron content is 2%, the corrosion resistance is the best. The minimum self-corrosion current density is 1.75766 × 10^−11^ mA·cm^2^, and the maximum self-corrosion potential is −0.254438 V. Maximum impedance curve radius. At this time, the wear resistance of the surfacing layer is also the best.

## 1. Introduction

With the rapid development of chemical industry, electric power, food, and other fields, surfacing alloy on the sealing surface of valve plays a wear-resistant role, which is more and more widely used in industrial fields. Adding boron to the flux cored wire can improve the wear resistance of the sealing surface of the valve, but the special application environment of the valve is relatively complex [1]. Therefore, the effect of boron addition on the corrosion resistance and wear resistance of surfacing layer and the transition coefficient of alloy elements should be considered when preparing flux cored wire. Therefore, the alloy powder added to flux cored wire should ensure the corrosion resistance of austenitic stainless steel. Ferroboron as an additive has the advantages of low price and low cost. The hardness and wear resistance of austenitic stainless steel are improved by adding ferroboron compound [2]. At present, while improving the wear resistance, it will affect the corrosion resistance of the surfacing layer. First, ensure that the corrosion resistance is not affected after adding ferroboron, and then improve the wear resistance of the austenitic stainless steel surfacing layer [3].

In order to study the action mechanism of boride in steel, a large number of researchers have carried out relevant research. Kocaman [4] et al. found that boride was formed in situ in the surfacing layer and existed in the form of eutectic boride phase, mainly by α(Fe Cr), tetragonal (Fe, Cr)_2_B, orthogonal (Fe, Cr)_2_B, and trace (Fe, Cr)_3_(C, B) phases. The surfacing alloy has good corrosion resistance. Medvedovski [5] et al. studied the high temperature corrosion of iron boride coating in geothermal power generation under different corrosion environments. The double layer boride-based coating has successfully withstood the effects of high temperature/high pressure water and steam, H_2_S, CO + CO_2_, chloride, and hydrocarbon in simulated conditions and strong acidic environment. Günen [6] et al. studied the surface boronizing treatment of aisih13 steel with mixed B_4_C and NaBF_4_ powder. The electrochemical corrosion (3.5 wt% NaCl) boronizing layer has a single-phase structure (Fe_2_B) at 800 °C and a two-phase structure (FeB + Fe_2_B) at higher boronizing temperatures (900 °C and 1000 °C). Under all boronizing conditions, the boronizing layer is dense and free of cracks. The corrosion resistance of boronized aisih13 steel is 33.5 times and 2.4 times higher than that of untreated aisih13 steel and martensitic aisi431 steel, respectively [7]. The effect of element B on the performance of austenitic stainless steel is rarely reported. Therefore, this paper takes element B as the research object to study its microstructure and performance under different B content.

## 2. Materials and Methods

Cr, Ni with purity greater than 99.9%, and ferroboron alloy powder with boron content of 18% were used in the test. The powder is provided by Xinke experimental supplies Co., Ltd. (Shenyang, China). The powder needs to be dried in advance to remove the water in the powder and ensure the purity. The non-dry flux cored wire easily produces defects such as pores and cracks.

The boron content of the alloy powder is 0%, 0.5%, 1%, 1.5%, and 2%, respectively, according to the pre-designed mass ratio. The alloy powder with different composition ratio is added into the steel strip and made by rolling and drawing Φ 2.4 mm flux cored wire. The steel strip is provided by Shanghai Yexin company (Shanghai, China), and the composition of steel strip is shown in Table 1. The 1 m long steel strip was weighed, and its weight was 136.33 g. The weight of one meter filled powder is 90.89 g, and the actual filling rate is 40% according to Formula (1). The surface of the base carbon steel is polished, the oxide scale is removed, cleaned with absolute ethanol, and then overlaid on the low carbon steel plate by means of MIG welding. Welding current 200 A, welding voltage 24 V, welding speed 10 cm/min, gas flow 12 L/min [5].
(1)$$\rho =\frac{m}{M+m}$$

In Formula (1): *ρ* is the filling rate, *M* is the quality of welding wire, *m* is the weight of the added powder. The quality of welding wire is *m + M*.

The experimental analysis instrument was provided by the material College of Shenyang University of technology. The phase structure of surfacing layer was analyzed by X-ray diffraction (XRD). The specific parameters are pure copper target, voltage 40 KV, current 30 mA, step size 2 °/min, scanning range 20° to 90°. The grain structure and phase composition were characterized by Geminisem 300 field emission scanning electron microscope equipped with energy spectrometer (EDS). The electrochemical corrosion properties of surfacing samples were tested by electrochemical workstation. Refer to electrochemical corrosion standard ASTM 5008. The electrochemical corrosion process in seawater environment was simulated. The corrosion medium was 3.5% NaCl solution. The weld overlay sample is cut into 10 mm × 10 mm × 20 mm by wire cutting, then the surface of the surfacing layer is polished with 200 #, 600 #, 800 #, 1000 #, 1500 #, and 2000 # sandpaper, and then the test piece is mechanically polished with water soluble diamond abrasive paste with a particle size of 1.5. Finally, the sample is sealed with epoxy resin to prevent the surface of the stacked welding layer from being corroded. The working parameters are as follows: the saturated calomel electrode is used as the reference electrode, the platinum electrode is used as the auxiliary electrode, and the scanning speed is 10 mV/min. Use HR-150A Rockwell hardness tester to measure the macro hardness of the surfacing layer. The sample size is 10 mm × 10 mm × 20 mm. Refer to T/CSTM 00646.2-2021 test standard. MMU-5G friction and wear tester were used to carry out wear tests at different temperatures on surfacing layers with different B additions. The sample size is Φ 3.9 mm × 15 mm. Before wear, the end of the sample shall be polished flat, the surface of the friction pair shall be polished and polished, and then ultrasonic cleaning and drying shall be carried out to remove impurities and oil stains remaining on the surface of the sample. The temperature is set at 200 °C, the friction and wear test time are 30 min, and the wear test specification parameters are shown in Table 2. After the test, weigh again with an electronic balance and record it as the weight after wear. The difference between the two weights is the weight loss of wear, as shown in Formula (2).
(2)$$\Delta m=m_{1}-m_{2}$$

In the formula: *m*_1_ is the mass before weighing; *m*_2_ is the mass after weighing; ∆*m* is the weight loss mass after calculation.

## 3. Experimental Results and Discussion

### 3.1. Effect of B Content on Phase Structure

In order to explore the effect of different B content on the phase structure of surfacing layer, JMatPro 10.0 software was used to simulate the phase structure of surfacing layer. The main method is to input each formula into the software in order to analyze the changes of various phases under different b contents, as shown in Figure 1. It can be seen from Figure 1 that M_3_B_2_ phase gradually increases with the increase of B element content. The corrosion resistance of fecrnibx coating prepared by laser cladding in 3.5% NaCl was studied according to Zhang et al. [6]. When 1.5 < x < 2.0, the corrosion resistance of the coating increases with the increase of B content. Therefore, it is most appropriate to add element B to 2%, and the corrosion resistance of the surfacing layer should be the best at this time. 

Figure 2 is the X-ray diffraction comparative analysis diagram of surfacing layers with different B content. The X-ray diffraction pattern of the sample is compared with the standard PDF card. The results show that when the content of B in the flux cored wire is 0, the microstructure of the surfacing layer is mainly Fe Ni Cr austenitic phase. With the addition of Fe_2_B, compared with the standard PDF card, it is found that the main body of the surfacing layer is Fe Ni Cr austenitic phase, M_3_B_2_ (M = Fe, Cr), and a small amount of Fe_3_(B, C) hard phase [7]. When the content of B in flux cored wire was 2.0%, the diffraction peak of M_3_B_2_ shifted slightly to the right. This is because the size of M_3_B_2_ phase is smaller at this time, which makes the performance of deposited metal more excellent [8].

The austenite grain size with different boron content is obtained by Jade’s treatment of the diffraction curve with the Scheler formula shown in Formula 3, as shown in Table 3. The grain size of boron free austenite is 180 ± 3 nm. Boride is formed with the addition of boron, and the formation temperature of boride is higher. During the cooling process of the molten pool, with boride as the core, austenite begins to nucleate, as shown in Figure 3. With the increase of B content, the compound formed with Fe element in the steel pinned the grain boundary, which strongly hindered the grain growth, and when B was 2%, the refinement effect of the structure was the best. The grain size of austenite is 119 ± 2.8 nm.
(3)$$D=\frac{K\lambda }{\beta cos\theta }$$

### 3.2. Effect of B Content on Microstructure

Figure 4 shows the microstructure of stainless steel alloy surfacing layer with different B content under scanning electron microscope. According to the XRD pattern, the alloy is composed of matrix Fe-Cr-Ni phase and boride. With the increase of boron content, the boride on the alloy matrix gradually increases. The boride is mainly dark gray long rod chromium rich boride and a small amount of bright white Mo rich reticular boride. Figure 4a shows the microstructure of the surfacing layer without B, and its matrix structure is austenitic structure (as shown by the arrow in Figure 4a γ). The microstructure is continuous and evenly distributed network dendrite. Compared with Figure 4a, there are obvious grid borides and rod borides in the Figure 4b,c structure of alloy with low B content, but they are relatively small and mainly distributed at the grain boundary [9]. Compared with alloys with higher B content, the microstructure is similar. The increase of boron content provides conditions for the growth of boride. Boride growth connects the network boride with the dispersed rod boride [10].

It can be seen from the energy spectrum analysis in Figure 5 and Table 4 that the atomic percentages of boron atoms are Fe and Cr, and it is further proved that the boride is mainly M_3_B_2_ boride. A large number of Cr atoms are enriched around borides. The formed Cr_2_O_3_ forms a passive film on the metal surface, which plays the role of passivation, and then improves the corrosion resistance of the surfacing layer [11].

### 3.3. Effect of B Content on Corrosion Resistance

Figure 6 is the comparison diagram of the electrochemical polarization curve of the addition amount of Boron on the surfacing layer in 3.5% NaCl solution. According to the curve, it can be seen that the alloy polarization curves of surfacing layer with different B addition are different, indicating that they have different corrosion resistance. Each component has been passivated, and the larger the passivation interval, the better the corrosion resistance of the material. Different boron contents have different icorr and OCP. It can be seen from Figure 6 that the icorr of surfacing layer decreases with the increase of B addition. Icorr is a dynamic criterion which directly reflects the corrosion rate of surfacing layer, and OCP is a thermodynamic criterion which directly reflects the trend of corrosion. The icorr and OCP under different boron additions are fitted by Tafel method. The fitting results are shown in Table 5. It can be seen from the table that with the increase of boron content, the icorr decreases and OCP increases gradually. When the B content is 1.5%, the boride changes from orthorhombic (Cr, Fe)_3_B_2_ to tetragonal (Fe, Cr)_2_B, which reduces the corrosion resistance of the coating [12]. When the amount of boron is 2%, the icorr is the smallest and the OCP is the highest, which are 1.75766 × 10^−11^ mA·cm^2^ and −0.254438 V respectively. The passivation zone is wide and the slower the corrosion rate is, the better the corrosion resistance is. The corrosion resistance increases with the addition of boride, which is mainly due to the gradual increase of boride M_2_B.

In order to study the effect of boron addition on the passivation characteristics of surfacing layer in 3.5% NaCl solution, AC impedance test was further carried out on the sample, and the results are shown in Figure 7.

According to [13], the larger the radius of the capacitive arc in the Nyquist plots diagram, the better the corrosion resistance of the surfacing layer. It can be seen from Figure 7 that the corrosion resistance of the arc surfacing layer is the same. When boron is not added, the capacitive reactance radius of the surfacing layer is the smallest. When the boron content is 2%, the capacitive reactance radius is the largest. The capacitive reactance radius of different boron content increases gradually with the increase of boron addition. The larger the capacitive reactance radius, the better the corrosion resistance. This shows that the charge transfer resistance of surfacing layer gradually increases with the increase of boron content, forming a more stable passive film.

It can be seen from the Bode diagram shown in Figure 8 that the trend of each curve is similar, which indicates that the pitting reaction mechanism of the sample is the same. It can be seen from the figure that with the increase of boron addition, the impedance value of low frequency increases, the phase angle gradually increases, and the horizontal frequency range of phase angle becomes wider. When the boron addition is 2%, the impedance value of surfacing layer is the largest, the phase angle is the largest, and the phase horizontal frequency range is the widest, which indicates that there is a denser passive film of surfacing layer when the boron addition is 2%.

For the surfacing layer with different boron addition, the equivalent circuit, as shown in Figure 9, is used to fit the EIS data [14,15], and the results are shown in Table 5. Where RS is the electrolyte solution resistance, CPE_1_ is the electric double-layer capacitance, Rt is the charge transfer resistance, CPE_2_ is the passive film capacitance, and Rf is the passive film resistance. The smaller the R_t_ value, the easier it is for ions to enter the electric double layer. The lower the R_f_ value, the weaker the protection of the passive film. The smaller the CPE_1_, the fewer the number of surface defects of the passive film. The lower the CPE_2_, the stronger the protection of the passive film [16,17]. It can be seen from Table 6 that the CPE_1_ and CPE_2_ values of surfacing layers with different boron additions are very small, which indicates that their passive films are very stable. The passivation film resistance R_f_ is greater than the charge transfer resistance R_t_, which indicates that the passivation film plays a major role in the corrosion resistance of the material, because when the boron addition is 2%, the CPE_1_ value is the smallest and the Rf value is the highest, which indicates that the material contains a complete passivation film and corrosion resistance.

### 3.4. B Effect on Mechanical Properties

Figure 10a,b, respectively, show the influence of the mass fraction of element B on the macro hardness and wear loss of the surfacing layer. The specific data are shown in Table 7. When the addition of B is 0%, the wear loss of surfacing layer is 0.0827, and the hardness is 18HRC. With the increase of the addition of B element, the wear amount decreases gradually. When the addition of B is 2%, the best match is achieved, the wear of the surfacing layer is 0.0141 g, and the hardness is 43.5 HRC. Compared with the surfacing layer without B, the wear resistance is increased by 82.9%, the hardness is increased by 58.6%, and the hardness and wear resistance of the surfacing layer are significantly improved. Although there is no inevitable linear relationship between hardness and wear resistance, the content of boride in surfacing layer is increased by increasing the content of B element in flux cored wire, which improves the hardness and wear resistance of surfacing [18].

Figure 11 shows the friction coefficient comparison of different boron additions. It can be seen from the figure that the order of the friction coefficient of the sample is B0 > B0.5 > B1 > B1.5 > B2.0. At the same time, with the increase of boron content, the pre-grinding period of friction coefficient gradually decreases. The pre-grinding period at 0% is about 350 s. After the pre-grinding period, the friction coefficient fluctuates around 0.41. The wear debris accumulated on the surface of surfacing layer has the effect of self-lubrication. A large amount of heat energy is generated in the friction process, which improves the temperature of the contact surface. The surfacing layer softens under high temperature conditions, so that the friction coefficient tends to be stable. At the same time, with the increase of boron content, the pre-grinding period of friction coefficient gradually decreases. When the boron content reaches 2%, the pre-grinding period is the shortest, and the pre-grinding period is about 120 s. 

As shown in Figure 12, the wear morphology is basically consistent with the wear curve. With the increase of boron content, the wear mechanism of surfacing layer gradually changes from abrasive wear to abrasive wear and a small amount of adhesive wear. In terms of wear degree, the surfacing alloy without boron belongs to austenitic stainless steel, with good plasticity and toughness and poor hardness [12]. Therefore, the wear is relatively serious, the furrow is relatively deep, and there are large pieces of massive spalling. The microstructure of surfacing alloy containing B is mainly austenite and M_2_B, Fe_3_(C, B) boride, so the wear resistance is significantly improved. With the increase of B content, rod boride gradually increases, which increases the wear resistance of surfacing alloy. The wear morphology shows that the furrow gradually becomes shallow, and the massive flakes gradually become smaller. The reason for furrow and block spalling is that at the beginning of the experiment, the sample and the friction pair have relative friction from contact to friction. Due to the rough surface of the sample and the friction pair, the micro convex parts of surface are in contact with each other. Under the action of external load and shear stress, the convex parts have relative friction and sliding and peel off from the matrix. One part of the debris falls off the grinding surface, and the other part does not leave the grinding surface. They are pressed into the grinding surface of the material under the action of upward stress, resulting in furrows and plastic deformation of the grinding surface under the action of tangential force in the process of friction and wear. During wear, the temperature of the contact surface increases, and the wear mechanism changes from abrasive wear to adhesive wear, so a part of the wear debris will be pressed and adhered to the wear surface. With the extension of wear time, the adhesion formed on the surface gradually increases and becomes larger, reaching a certain volume and forming a block to fall off [19].

## 4. Conclusions

In this test, the alloy surfacing layer is prepared on the surface of low carbon steel by gas metal arc welding. The following conclusions are drawn:

(1) The surfacing alloy is mainly Fe Ni Cr solid solution, which forms boride and gradually increases with the increase of B content. The boron containing phase is mainly M_2_B and a small amount of Fe_3_(B, C). Its structure is mainly rod-shaped boride and a small amount of flake boride.

(2) When the boron content is 2%, the minimum self-corrosion current density is 1.75766 × 10^−11^ mA·cm^2^, and the maximum self-corrosion potential is −0.254438 V. The mechanism of boride improving corrosion resistance is that chromium rich (Cr, Fe)_2_B has stable corrosion resistance, and the enriched chromium forms Cr_2_O_3_, which improves the corrosion resistance of the stack layer.

(3) With the addition of boron, the hardness of surfacing layer is significantly improved. When the boron content is 2%, the wear resistance of surfacing layer is the best. Wear machine causes abrasive wear to adhesive wear, and the furrow gradually changes from deep to shallow.

## Figures and Tables

**Figure 1 materials-15-05884-f001:**
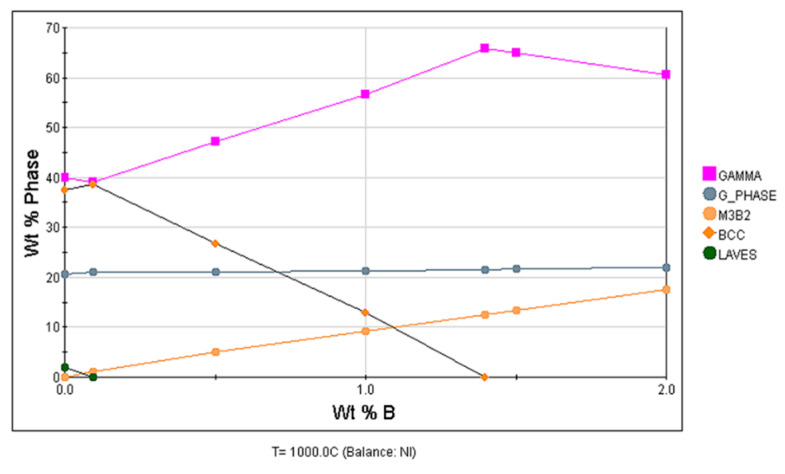
Effect of different content of B element on phase separation of austenitic surfacing.

**Figure 2 materials-15-05884-f002:**
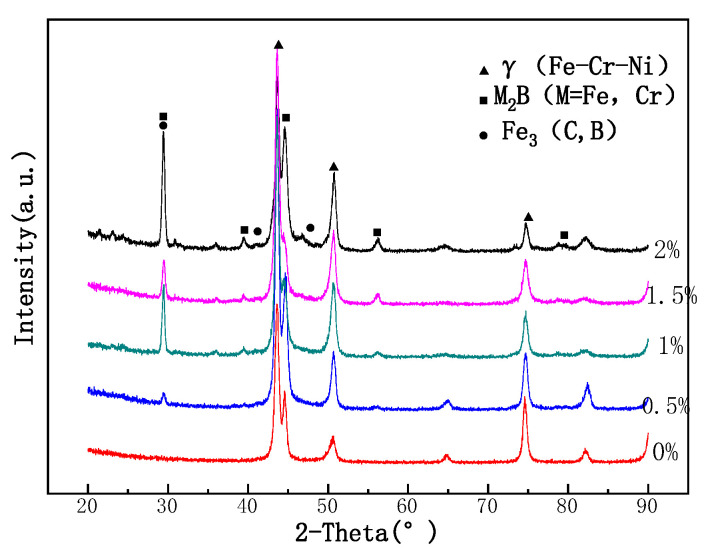
X-ray diffraction comparative analysis of different boron content in surfacing alloy.

**Figure 3 materials-15-05884-f003:**
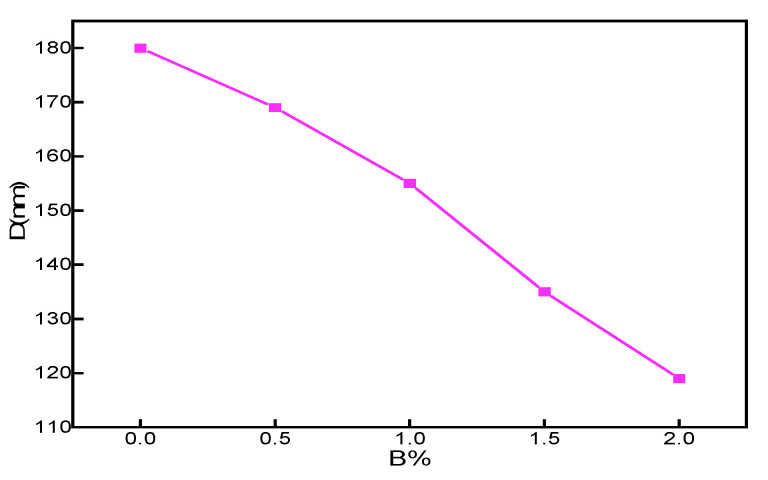
Grain size diagram of different nitrogen content.

**Figure 4 materials-15-05884-f004:**
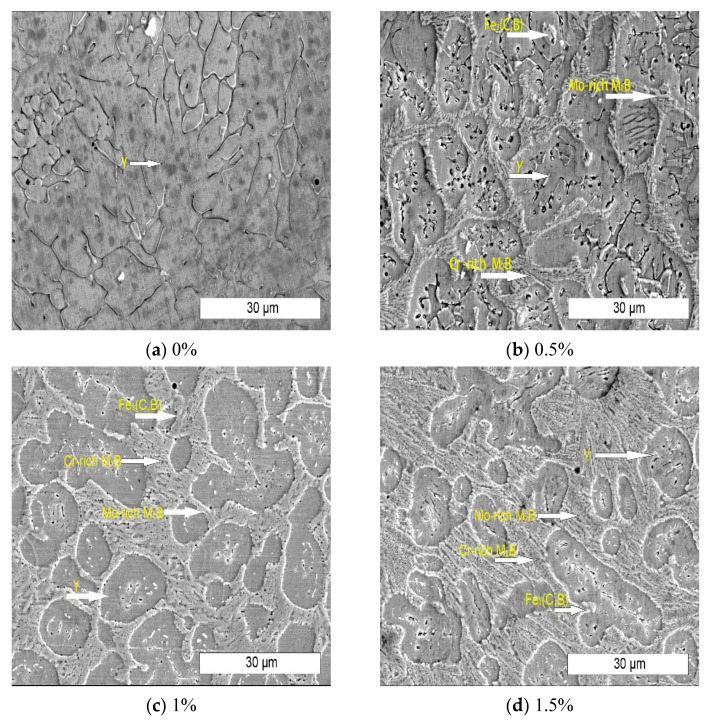
The microstructure of surfacing alloy with different B content.

**Figure 5 materials-15-05884-f005:**
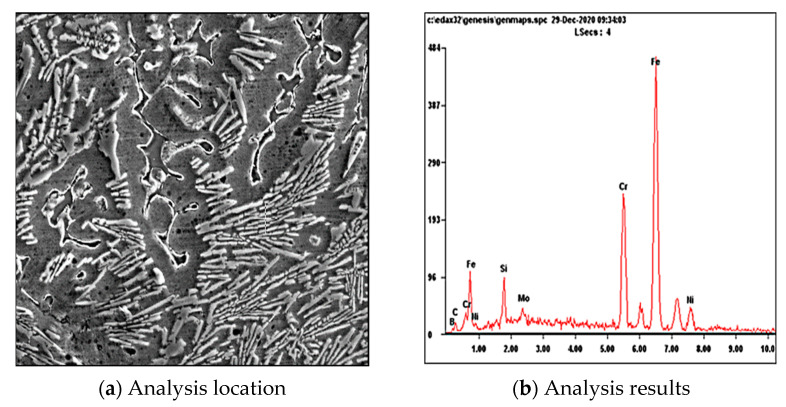
Boride energy spectrum analysis.

**Figure 6 materials-15-05884-f006:**
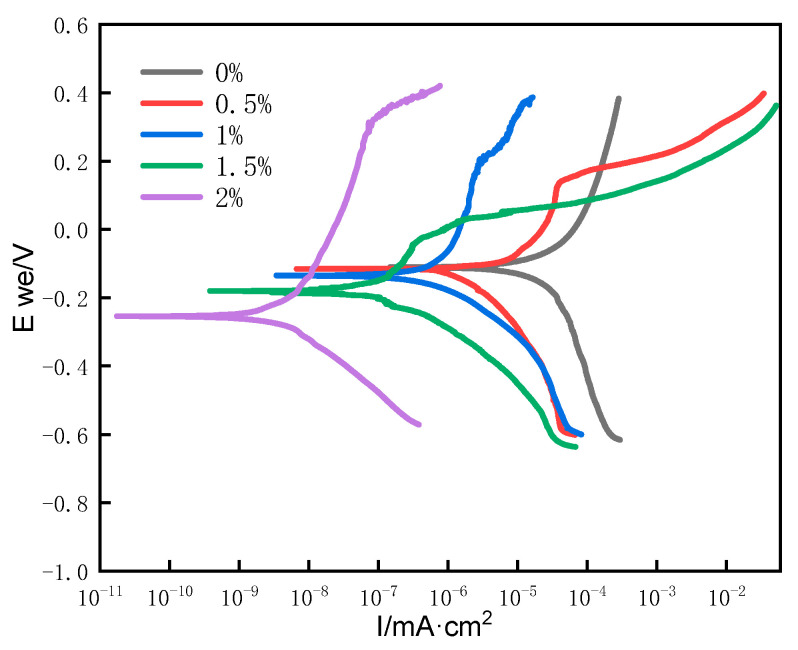
Comparison of polarization curves of surfacing layers with different boron additions.

**Figure 7 materials-15-05884-f007:**
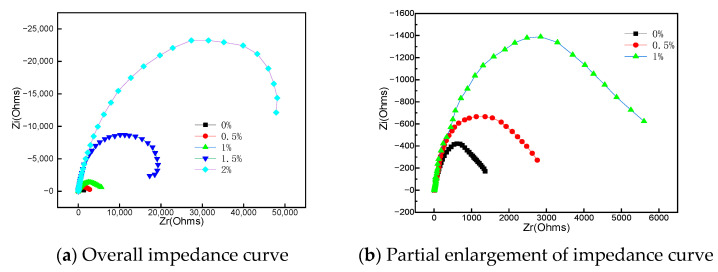
EIS in 3.5% NaCl solution: Nyquist.

**Figure 8 materials-15-05884-f008:**
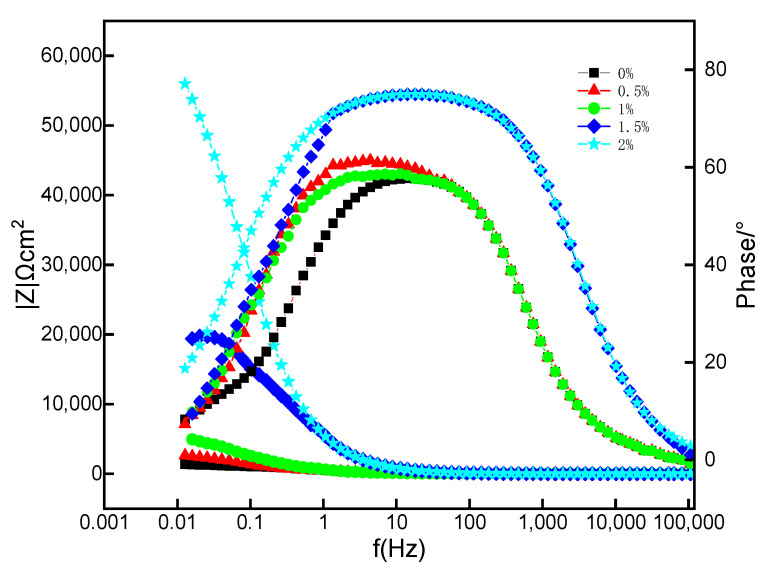
EIS in 3.5% NaCl solution: Bode plots.

**Figure 9 materials-15-05884-f009:**
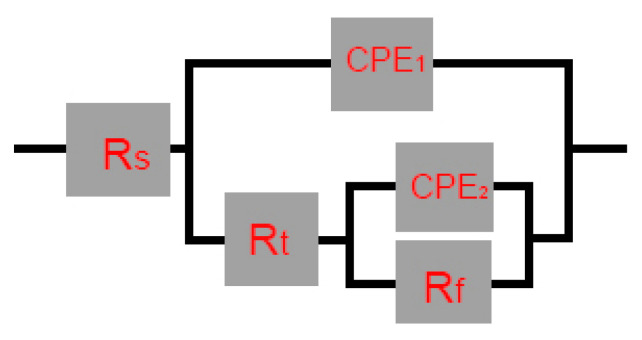
Electrochemical impedance equivalent circuit diagram of surfacing layer in 3.5% NaCl solution.

**Figure 10 materials-15-05884-f010:**
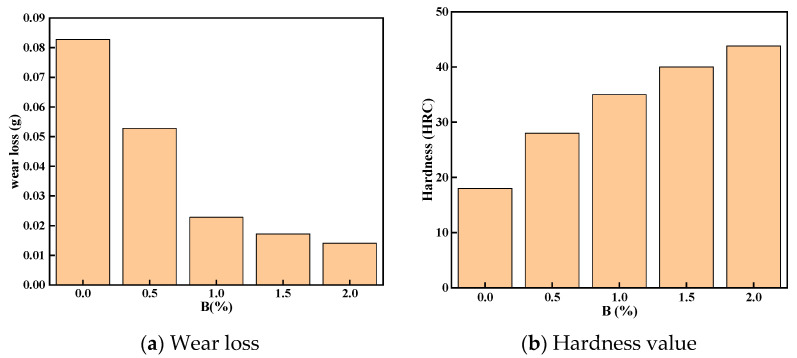
Wear loss and hardness value of surfacing layer with different amount of boron.

**Figure 11 materials-15-05884-f011:**
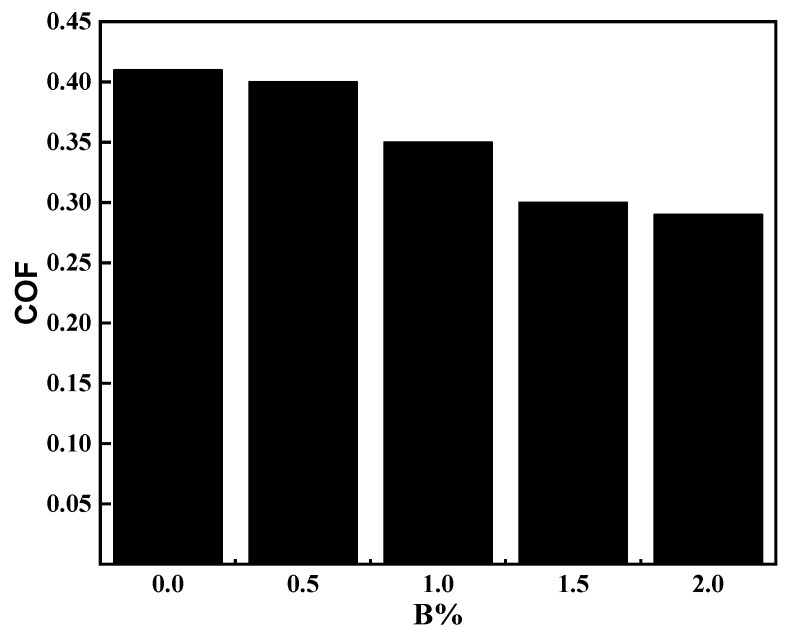
Comparison of friction coefficients of surfacing layers with different amounts of boron.

**Figure 12 materials-15-05884-f012:**
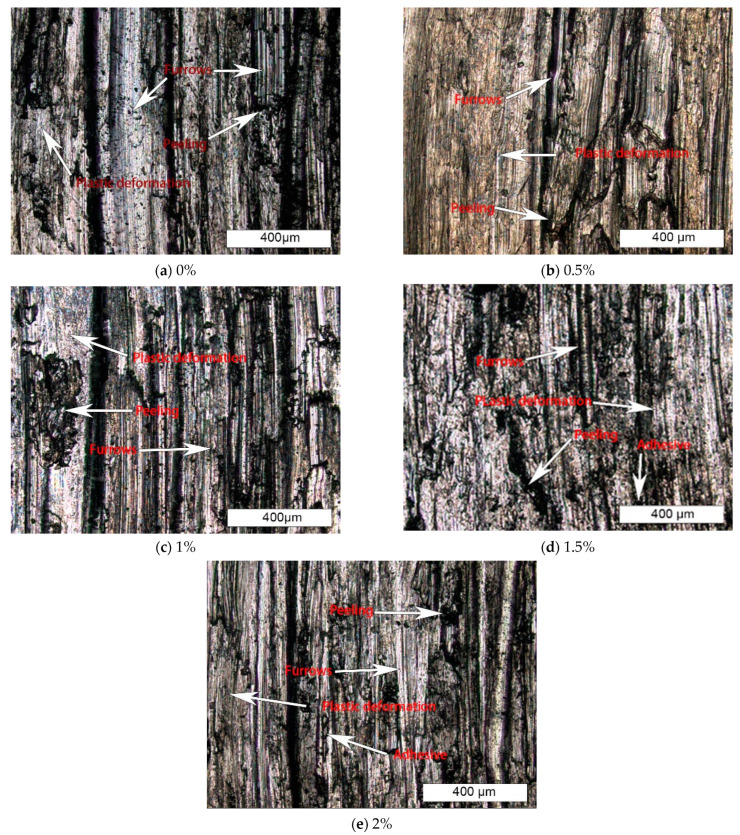
Wear morphology of surfacing layers with different amounts of boron.

**Table 1 materials-15-05884-t001:** Composition of 304 steel strip (mass fraction, wt%).

Element	C	Si	Mn	Ni	Cr	Fe
Content	<0.08	≤1	≤2	8	18	Bal

**Table 2 materials-15-05884-t002:** Friction and wear parameters.

Experimental Force (N)	Wear Time (min)	Friction Torque (N·m)	Revolutions (r/min)
200	30	10	100

**Table 3 materials-15-05884-t003:** Austenite grain size with different boron content (nm).

Element content	0	0.5	1	1.5	2
grain size	180	169	155	135	119

**Table 4 materials-15-05884-t004:** Energy spectrum composition of boride.

Component	B	C	Cr	Ni	Mo	Si	Fe
At%	29.41	3	26.35	4.48	0.41	3.71	32.44
Wt%	8.56	1.5	39.83	7.94	1.76	3.15	53.04

**Table 5 materials-15-05884-t005:** Electrochemical corrosion results of surfacing layer with different boron content.

B Addition (%)	Self Corrosion Potential (V)	Self Corrosion Current (mA·cm^2^)
0	−0.111	1.47 × 10^−7^
0.5	−0.115873	6.67054 × 10^−9^
1	−0.135169	3.3419 × 10^−9^
1.5	−0.18051	3.80451 × 10^−10^
2	−0.254438	1.75766 × 10^−11^

**Table 6 materials-15-05884-t006:** Fitting results of equivalent circuit of surfacing layer in 3.5% NaCl solution.

Sample	R_s_ (Ωcm^2^)	CPE1 (Ω-1s-ncm^2^)	N1	R_t_ (Ωcm^2)^	CPE2 (Ω-1s-ncm^2^)	N2	R_f_ (Ωcm^2^)
A	6.556	3.983 × 10^−4^	0.7378	1.221 × 10^3^	2.123 × 10^−2^	0.766	1.085 × 10^2^
B	6.683	2.913 × 10^−4^	0.7824	3.023 × 10^3^	3.905 × 10^−4^	0.7226	2.285 × 10^3^
C	6.627	3.253 × 10^−4^	0.7643	4.584 × 10^2^	3.678 × 10^−5^	1	4.377 × 10^3^
D	6.02	3.61 × 10^−5^	0.8474	1.825 × 10^4^	4.341 × 10^−4^	0.3310	5.19 × 10^3^
E	6.016	3.59 × 10^−5^	0.8	2.161 × 10^2^	1.89 × 10^−5^	0.8471	7.7 × 10^3^

**Table 7 materials-15-05884-t007:** Hardness and wear of surfacing layer under different boron additions.

Addition Amount	0	0.5	1	1.5	2
Average hardness	18	28	35	40	43.5
wear loss	0.0827	0.0528	0.0228	0.0172	0.0141

## Data Availability

Authors can confirm that all relevant data are included in the article.

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
