# Peer review of "Effect of B on Microstructure and Properties of Surfacing Layer of Austenitic Stainless Steel Flux Cored Wire"

_materials, 2022, doi:10.3390/ma15175884_

Round 1

Reviewer 1 Report

Dear Authors,

Kindly change the phrases in the line no 12-15. Some of the lines are poorly constructed. I request the authors to verify it

When the boron content is 2%, the corrosion resistance is 15 the best. - Line no 15 & same is repeated in line no 19

The no of references used in the study is very limited.

Line no 22 to 36 can split with more references.

The following line no 57-58 is confusing ( The powder needs to be dried in advance, because the 57 powder is placed for a long time to absorb water.)

on what basis the authors choosen the powder filling rate is 40%

did the authors measured the phase analysis of the powder

The ASTM test details of electrochemical set up is missing

JMatPro software version details not available. Where is the input parameter. Fig 1 is not clear. The chinese word in the top of the figure can be replaced

The marginal amt of peak shifting occured in 2%B in XRD. the details are not explained properly

The sample details or the photograph information is not clearly defined in the present format.

What is the validity and error values in the Jade's treatment with the use of Scheler formula?

It is advisable to keep the machine information in SEM images. Did the authors observed the image in SE or BSE mode? The yellow color word can be change . The magnification details are not available any where in the manuscript.

The OCP details are not mentioned in the corrosion plot.  The Corroison resistance details are not mentioned in details. The discussion need to improve.

over all I did not find any novelty in the work.

Author Response

Thank the reviewers for their comments, and reply in word. Thank you for reviewing

Reviewer 2 Report

Effect of boron was studied to improve the corrosion resistance.

Interesting research has been conducted onthe improvement of corrosion resistance of iron-based materials.

But, some revision is required.

1. In 2. Materials and Method, a clear source statement is required for the materials and analysis apparatuses.

2. In Fig. 1, it is required the deletion of other languages, except the English.

3. the row 99, the author was used the standard PDF card, but the references are also required from other papers, books.. and so on.

4. As the formula 1, the author claimed that the grain size would be decresed. Further discussion of "why" is required.

5. For the surface properties, in Fig. 4, the author was examined the chemical analysis by using energy spectrum. For accurate analysis, XPS or SIMS analysis is recommended.

6. In Fig. 6, when the author used the 1.5 wt.%, corrosion resistance was rapidly decreased, especially pitting corrosion, why?

And, why did it suddenly increase in 2 wt.%?

7. The Nyquist plot and potentio-dynamic graph don't seem to be related.

8. What type of the reference and counter electrode were used?

9. It is recommended to analyze the electric circuit by comparing it with the FE-SEM cross-sectional images. And with discussion.

10. The caption of Figures should be revised to more accurately.

11. More accurate wear resistance test conditions are required, such as wear time(as the standard, ISO, ASTM..), counter ball, temperature..

12. The paper seems to be focused on simply listing the experimental results. The author should be revised to supplementary discussion and English proofreading.

Author Response

(The authors gave the same response as above.)

Round 2

Reviewer 1 Report

The authors made a significant changes in the manuscript. However, the figure have the red color font ( Fig 11 ) is difficult to read. If possible, kindly replace it

Author Response

Thanks for the comments of the reviewer. The revision has been completed. Please review it

Reviewer 2 Report

All of comments was properly reflected.

But, Fig. 10 still inserts the other language, except English.

Author Response

(The authors gave the same response as above.)
